# Nutrient-Derived Beneficial for Blood Pressure Dietary Pattern Associated with Hypertension Prevention and Control: Based on China Nutrition and Health Surveillance 2015–2017

**DOI:** 10.3390/nu14153108

**Published:** 2022-07-28

**Authors:** Yuxiang Yang, Dongmei Yu, Wei Piao, Kun Huang, Liyun Zhao

**Affiliations:** NHC Key Laboratory of Trace Element Nutrition, National Institute for Nutrition and Health, Chinese Center for Disease Control and Prevention, Beijing 100050, China; yxyang_ninhccdc@126.com (Y.Y.); yudm@ninh.chinacdc.cn (D.Y.); piaowei@ninh.chinacdc.cn (W.P.); 15550807252@163.com (K.H.)

**Keywords:** hypertension, reduced-rank regression, dietary pattern, DASH diet

## Abstract

Background: Greater adherence of Dietary Approach to Stop Hypertension (DASH) or the Mediterranean dietary pattern were reported to be beneficial for blood pressure. However, both were established based on Western populations. Our current study aimed to explore a dietary pattern which might be suitable for hypertension prevention and control among Chinese adults nationwide. Methods: A total of 61,747 Chinese adults aged over 18 years from China Nutrition and Health Surveillance 2015–2017 was included in this study. Using reduced-rank regression (RRR) method, a dietary pattern with higher intakes of those nutrients which are inversely associated with the risk of hypertension was identified. DASH-score was also calculated for each participant for further validate the dietary pattern derived by RRR method. Multi-adjustment logistic regression was applied to examine the association between above two dietary patterns and hypertension prevention and control. Results: Dietary pattern named Beneficial for Blood Pressure (BBP) diet was characterized by higher fresh vegetables and fruits, mushrooms/edible fungi, dairy products, seaweeds, fresh eggs, nuts and seeds, legumes and related products, aquatic products, coarse cereals, and less refined grains and alcohol consumption. After multiple adjustment, protective effects showed on both hypertension prevention and control (for prevention: Q5 vs. Q1, OR = 0.842, 95% CI = 0.791–0.896; for control: Q5 vs. Q1, OR = 0.762, 95% CI = 0.629–0.924). For the DASH-diet, significant results were also observed (for prevention: Q5 vs. Q1, OR = 0.912, 95% CI = 0.854–0.973; for control: Q5 vs. Q1, OR = 0.76, 95% CI = 0.616–0.938). Conclusions: BBP-diet derived from Chinese adults has high conformity with the DASH-diet, and it might serve as an adjuvant method for both hypertension prevention and control.

## 1. Introduction

Globally, hypertension (HTN) prevalence is on the rise. It is estimated that adults with HTN will account for over 30% global adult population by 2025 [1,2]. HTN has the linkage with a wild-spectrum non-communicable chronic disease (NCDs), including cardiovascular diseases (CVD), renal impairment, cerebral vascular disease, and all-cause mortality [3,4]. However, it is commonly undetected as it does not present obvious structural and functional symptoms in the early stage, especially in the case of healthcare or screening service is limited [4,5]. In China, it is estimated that the prevalence of HTN had markedly risen from 15.7% in 1991 to 23.3% in 2015 among Chinese adults but remains inadequately controlled [6,7]. It has become a national public health priority in China, and there is an urgent need to explore broad-based, more effective, and affordable strategies for the prevention, screening, and treatment of HTN [6].

Many lifestyle behaviors are related to HTN prevention, such as a healthy diet, weight management, physical activity, control of smoking, excessive alcohol consumption, etc. [8,9,10]. Among them, the diet has been recognized as a critical factor in developing of HTN [11]. Dietary Approach to Stop Hypertension (DASH) diet has been widely considered a protective strategy for blood pressure control and further benefits for cardiac structure and function [12,13]. Meanwhile, the Mediterranean dietary pattern plays a role in blood pressure reduction by improving vascular function and reducing inflammation [14]. Of the established dietary patterns mentioned above, they are characterized by higher consumption of fresh vegetables and fruits, whole grains, legumes, nuts, and lower consumption of refined grains, red and processed meats, and sweets [13,14]. However, both DASH and Mediterranean diets were established based on Western populations [15,16] and may not fit well in representing and measuring Chinese dietary characteristics [17,18].

Little prior research has focused on the protective dietary pattern and HTN among the Chinese nationwide population. Research showed that a dietary pattern named “traditional Southern pattern” with higher consumption of fresh vegetables and fruits, poultry, aquatic products, and nuts has an inverse association with the risk of HTN [19]. However, this dietary pattern was also characterized by higher consumption of refined grain and meat, which may not consist with the previous studies [9]. Besides, this research did not include the Chinese elderly, an important part of the whole population. Meanwhile, the Reduced-Rank Regression (RRR), a method recently applied in nutritional epidemiology, could either identify the dietary pattern or explain the maximum variation of the response variables (such as nutrients) linked to the aim disease [20,21]. This study aims to explore the dietary pattern among all-adults in China, examine the relationship between DASH-score and current dietary patterns and the risk of HTN based on the data from China Nutrition and Health Surveillance (CNHS) 2015–2017. Moreover, we also examined the association between the dietary pattern and the condition of HTN control further to support the anti-HTN effect of the dietary pattern.

## 2. Materials and Methods

### 2.1. Study Population

The cross-sectional data were obtained from the CNHS 2015–2017, this survey conducted among Chinese people enrolled in 2015–2017 and supported by the Ethics Committee of the Chinese Center for Disease Control and Prevention (approval number: 201519-B). A stratified, multistage, and random sampling method was applied to include a representative sample from 31 provinces/ municipalities/ autonomous in China mainland. The aim of this survey was to obtain the information of dietary intake, lifestyle habits, medical examination, as well as disease condition and related risk factors among all-age Chinese people, further information about CNHS is available elsewhere [22]. At the beginning of the study, we included 78,480 participants in total. Those participants were excluded as follow process (Figure 1): incomplete dietary survey (*n* = 3118), incomplete basic information (*n* = 900), incomplete physical examination and laboratory test (*n* = 5719), implausible dietary energy intake < 500 Kcal/d or > 5000 Kcal/d (*n* = 2157), and pre-diagnosed coronary heart disease or stroke (*n* = 4839). Finally, 61,747 participants were included in this study, and all of them had signed the informed consent before the survey.

### 2.2. Basic Information Interview

Well-trained health investigators collected all the basic information, including socioeconomic status and lifestyle behavior. To avoid incorrect filling and improve the authenticity and accuracy of the survey, the whole survey was conducted face-to-face and recorded by the investigators. After that, multi-level quality control was carried out by professional stuff from the national and local Center for Disease Control and Prevention (CDC).

### 2.3. Medical Examination and Laboratory Test

Participants were required to finish medical examination on an empty stomach in the morning. To control measurement bias, unified standardized equipment, and machines of the same type and brand were used in all the monitoring locations. Two well-trained local CDC staffs measured height (accurate to 0.1 cm), body weight (accurate to 0.1 kg), and blood pressure (accurate to 1 mmHg) together. For laboratory test, 8 mL of overnight fasting blood was used to test all the blood indices, i.e., fasting glucose, total cholesterol, triglyceride, Low-Density Lipoprotein (LDL-C), High-Density Lipoprotein (HDL-C), serum uric acid, and glycohemoglobin. All the laboratory process was taken by professionals with strict quality control.

### 2.4. Dietary Assessment

A validated semi-quantitative Food Frequency Questionnaire (FFQ) was applied to reflect dietary habits during the past 12 months in CNHS 2015–2017 [23]. Sixty-four kinds of food, including staple foods, vegetables, fruits, animal and aquatic products, eggs, legumes and mixed beans, dairy products, soft and alcoholic beverages, etc., were surveyed and integrated into 28 food items in current study. The daily consuming weight of the above food items was aggregated based on their daily/weekly/monthly/yearly consuming frequency and consumption in a single time. To estimate daily dietary energy and nutrients intake more accurately, according to the monthly consumption records of the edible oil and condiments and the frequency of family members eating at home during the last month, the average daily intake of edible oil and other condiments was calculated. The daily intake of energy and nutrients per participant was calculated based on the China Food Composition Table (2009 & 2018) [24,25]. While the parts supplied from nutrient supplements were omitted.

### 2.5. Definition of HTN and Other NCDs

Systolic Blood Pressure (SBP) and Diastolic Blood Pressure (DBP) were obtained due to the average value of the three-time measurement. HTN was defined as SBP ≥ 140 mmHg and (or) DBP ≥ 90 mmHg and (or) taking anti-HTN medication within two weeks [26]. Our study included other health outcomes, such as Diabetes Mellitus (DM) and hyperlipidemic. In addition to participants who had been diagnosed by a township health center/community health service center/higher-level medical institute or taken related-disease medical treatment, we also defined participants whose serum total cholesterol ≥ 6.2 mmol/L and (or) triglyceride ≥ 2.26 mmol/L and (or) LDL-C ≥ 4.14 mmol/L and (or) HDL-C ≤ 1.04 mmol/L as hyperlipidemic [27], and participants whose fasting plasma glucose ≥ 7.0 mmol/L and (or) glycohemoglobin level ≥ 6.5% were defined as DM [27]. Moreover, well-controlled HTN was defined as participants who had been previously diagnosed with HTN and reported taking anti-HTN medication within two weeks whose current SBP < 140 mmHg and DBP < 90 mmHg [26,28].

### 2.6. Dietary Pattern Analysis

RRR is a recently emerging method applied in dietary pattern analysis [20]. Unlike the traditional dietary pattern analysis, RRR could combine the advantage of both the priori and posterior methods. It selects response variables (such as nutrients and biomarkers) based on the prior knowledge as the intermediate variables between the food groups and target outcome. It could recognize the food groups with collinearity to maximumly explain the variance of response variables and examine the mechanistic pathway from diet to aim outcomes [29,30].

Nutrients including thiamin, riboflavin, vitamin C, Calcium, Magnesium, Potassium, and dietary fiber from diet have been reported to have protective effects on the risk of HTN due to either genetic or metabolic mechanisms [31,32,33,34,35,36,37,38,39,40]. Before RRR analysis, we conducted a preliminary analysis of our study population, and all these nutrients were inversely associated with blood pressure. Thus, we selected the above 7 nutrients as response variables and 28 food items (i.e., refined grains, coarse cereals, legumes and related products, nuts and seeds, tubers, fried staples, western staples, fresh vegetables, dried vegetables, fermented vegetables, mushrooms/edible fungi, seaweeds, fresh fruits, dried fruits, dairy products, pork, beef/lamb and others, poultries, processed meats, animal organs, aquatic products, fresh eggs, processed eggs, snacks, 100% vegetable and fruit juice, coffee, sweetened beverages, and alcohol) as explanatory variables after energy adjustment [41]. The first dietary pattern was retained for further analysis since it explained the largest variation of the response variables [29]. A food item with absolute factor loading over 0.1 was used to describe the characteristics of the dietary pattern. Higher absolute factor loading of food items reflects higher consumption, and the dietary score was calculated by standardized food intakes and their weight within the dietary pattern [21]. Participants were afterward divided into quintile groups due to dietary scores to conduct further analysis.

Moreover, the DASH-score was also applied in the study to evaluate its relationship with the risk of HTN among the participants in CNHS 2015–2017 and test whether the derived dietary pattern had conformity with it since the DASH diet is widely reported to have benefits for blood pressure control [42]. DASH-score was calculated by gender and the quintile of eight food items’ daily intake, five for positive points including whole grains, fruits, vegetables, nuts and legumes, whole grains, and all dairy products (due to low consumption of low-fat dairy products in Chinese population, we applied all dairy products to calculate this components) [43], and three for negative points including sodium, red & processed meats, and sugar-sweetened beverages. Further information on DASH-score is available in the previous study [44]. Participants were also divided into quintile groups due to DASH-score to conduct further analysis.

### 2.7. Covariates

Covariables are used for character description and multiple adjustments in Logistic models and are as follows. (1) Age was further divided into 18~<30, 30~<45, 45~<60, ≥60 years. (2) Body Mass Index (BMI) was divided into underweight (<18.5), normal (18.5~<24), overweight (24~<28), and obese (≥28) according to the China Working Group on Obesity [26]. (3) Living areas were divided into urban and rural. (4) Education level was divided into primary school or below, junior middle school, and high school or higher. (5) Marital status was divided into married and other status (such as live alone, divorced, or live with others). (6) Income (per-capita of household within a year) was divided into not given, low (<10,000 CNY), medium (10,000~<25,000 CNY), and high (≥25,000 CNY). (7) Physical activity was divided into low, medium, and high according to weekly total Metabolic Equivalent (MET-min/week) and total duration of different physical activities [45]. (8) Sedentary behavior was divided into <2, 2~3, and ≥4 h per day. (9) Sleeping duration was divided into <7, 7~8, and ≥9 h per day. (10) Smoke was divided into current smokers and non-smokers (including never smoked and already quit smoke). (11) Excessive alcohol drinking was divided into yes (equivalent to daily pure alcohol consumption over 25 g for males and 15 g for females) and no (for others) [46]. (12) Secondhand smoking was divided into yes (usually, being exposed to secondhand smoke more than half a week) and no (for others). (13) Medical examination status was divided into yes (had received within one year) and no (for others). (14) Family history of HTN (Yes/No) was deemed as any one of lineal relative (including grandparents, parents, or siblings) used to be diagnosed with HTN.

### 2.8. Statistical Analysis

Continuous variables with normal distribution were described with mean and standard error, while else were described with median and the 25th and 75th percentile and compared by student t-test and Wilcoxon test between different groups, respectively. Categorical variables were described with the amount and related proportion (%) and compared by chi-square test between different groups. PROC PLS was used for conducting RRR analysis and calculating related diet score for each participant [20]. Multiple adjustment logistic regression to compare the effect of dietary patterns on HTN and well-controlled HTN. Model I is the crude model without adjusting any covariates, model II is adjusted by gender, age, and BMI, and model III is further adjusted by remaining covariables.

Subgroup analysis was conducted to test the interaction between dietary pattern score and risk of HTN with gender (male/female), age (18~<30, 30~<45, 45~<60, ≥ 60 years), physical activity (low, medium, high), current smoker (yes/no), DM (yes/no), and hyperlipidemic (yes/no) by comparing the highest quintile group with the lowest in full-adjustment logistic regression.

Furthermore, sensitivity analysis was also conducted to robust our results. First, we further adjusted DM and hyperlipidemia in model III to examine the association between dietary scores and HTN. Then, we used completed participants by excluding those did not report household income and were pre-diagnosed with HTN to perform multiple-adjustment logistic regression.

SAS v. 9.4 (SAS Institute Inc., Cary, NC, USA) was used for all the statistical analyses in this study, of all the analyses, statistical significance was defined as a two-sided *p* value less than 0.05.

## 3. Results

### 3.1. Dietary Pattern Extracted by RRR

The first dietary pattern explained 66.08% variation of response and 5.73% variation of food items, characterized by higher fresh vegetables and fruits, mushrooms/edible fungi, dairy products, seaweeds, fresh eggs, dried fruits, nuts and seeds, legumes and related products, aquatic products, coarse cereals, western staples, less refined grains, and alcohol consumption (Figure 2). For each response variable, the dietary pattern explained variation varies from 81.05% (for Potassium) to 25.29% (for thiamin), and all of them had a positive correlation with the dietary score. Therefore, we initially named the first dietary pattern (DP1) derived by RRR method as diet beneficial for blood pressure (BBP-diet) for further step of analysis. Further information is available in Appendix A.

### 3.2. Characteristics of Participants in Quintile Groups

A total of 61,747 participants were included in current study. Statistical significance was observed when compared different variables between quintile groups divided by BBP-diet score. As the results shown in Table 1, those participants in the highest quintile had lower systolic and diastolic blood pressure. They were more likely to be female, live in the urban areas, have higher socioeconomic levels, adequate sleeping duration, higher DASH-score, and received medical examinations within one year, but they also have higher BMI and longer sedentary behavior. Meanwhile, the highest quintile had fewer participants who were current smokers, exposed to second-hand smoke, and excessive alcohol drinking. As for outcomes, compared to the lowest quintile, there was fewer patients with HTN, but more proportion of DM, hyperlipidemic, and those who had family history of HTN in the highest quintile.

### 3.3. Food and Nutrients Daily Intake of Participants in Quintile Groups

For daily food intake, participants with higher BBP-diet score were more likely to consume those foods with higher factor loadings (≥0.1). They consumed less refined grains and pork, the food item with negative factor loadings. Interestingly, although some food items with higher absolute factor loadings, their median values were all equal or near to zero, such as dried fruits, western staples, and alcohol, making it hard to compare actual intakes between different quintile groups. As for daily nutrient intake, except for total energy, fat, carbohydrate, and sodium, all the other nutrients achieved significantly higher intakes in the highest quintile. Further information is available in Appendix A.

### 3.4. Association between Dietary Scores and Risk of HTN

These results were shown in Table 2. In the full-adjustment logistic model, the BBP-diet from RRR showed a negative association with HTN (Q5 vs. Q1, OR = 0.842, 95% CI = 0.791–0.896, *p*-trend < 0.0001). For DASH-score, a negative association between DASH-diet and HTN also showed in the full-adjustment logistic model (Q5 vs. Q1, OR = 0.912, 95% CI = 0.854–0.973, *p*-trend = 0.0063).

### 3.5. Subgroup Analysis

The results of evaluating the association between BBP-diet scores and HTN in subgroups were shown in Figure 3, and further information is available in Appendix A. The protective effect on HTN was observed regardless of gender, the level of physical activity, smoking status, and with hyperlipidemic or not, but the higher protective effects still showed among those who were males (males: OR = 0.819 vs. females: OR = 0.906, *p* for interaction = 0.7001). Moreover, significant results only showed in participants aged over 45 years (45~<60 years: OR = 0.881; ≥60 years: OR = 0.784), while potential effect modified by age group was not observed (*p* for interaction = 0.1254). For the subgroups divided by DM, the protective effect of the BBP-diet on HTN tended to be more pronounced in participants who were DM patients (*p* for interaction < 0.0001).

### 3.6. Sensitivity Analysis

Results of sensitivity analysis were shown in Appendix A. After further adjusting DM and hyperlipidemic in model III, both BBP-diet and DASH-diet still had significant effects on HTN prevention. After excluding participants who did not report household income or were pre-diagnosed with HTN, the same results were also observed with only slight changes.

### 3.7. Association between Dietary Scores and Well-Controlled HTN

Moreover, among those HTN patients who received medical treatment within two weeks, the association between dietary scores and well-controlled HTN was also analyzed. The results were shown in Table 3. For the BBP-diet derived from RRR, patients in the highest quintile tended to have a higher probability of getting well-controlled (Q5 vs. Q1, OR = 0.762, 95% CI = 0.629–0.924, *p*-trend = 0.002). The DASH diet was also observed to have protective effects on HTN control (Q5 vs. Q1, OR = 0.76, 95% CI = 0.616–0.938, *p*-trend = 0.009).

## 4. Discussion

From this nationwide cross-sectional study, we identified the dietary pattern by RRR method which was named as BBP-diet. This pattern was characterized by higher fresh vegetables and fruits, mushrooms/edible fungi, dairy products, seaweeds, fresh eggs, dried fruits, nuts and seeds, legumes and related products, aquatic products, coarse cereals, western staples, and less refined grains and alcohol consumption. It positively correlated with the intakes of thiamin, riboflavin, vitamin C, Calcium, Magnesium, Potassium, and dietary fiber which were selected as the response variables could serve as protective factors in HTN. After multiple adjustments, a higher BBP-diet score was found a protective effects on both risk for HTN and its control, which indicated its potential role for HTN prevention and management in the further applications.

The BBP-diet has high conformity with DASH-diet and Mediterranean diet which were widely reported to possibly be beneficial for cardiovascular health [14,42,47], all of which were represented by limited calories and sodium intake, higher consumption of vegetables, fruits, nuts, legumes, coarse cereals, aquatic products, less refined grains, and red meat. Moreover, we also identified extra food items which also showed relative higher consumption within BBP-diet, including mushrooms/edible fungi, seaweeds, and fresh eggs. Although dried fruits, western staples, and alcohol also had higher absolute factor loadings, the median daily intake did not show an obvious difference between quintile groups. Thus, it is hard to reflect the significant dietary difference by the intake of dried fruits and western staples among our samples. However, when we considered the condition of excess alcohol drinking, which is one of the risk factors for HTN [4], the results showed there were less participants who drank alcohol excessively with the increase of BBP-diet score.

In some countries like China, Japan, and Korea, due to the flavor and general consideration of health effects, bacteria and algae foods are widely consumed in daily diet, and they are rich in several important vitamins, minerals, as well as dietary fiber which are essential for human health [48,49]. As previous research reported, a variety of mushrooms and seaweeds contain anti-HTN bio-components which could regulate the Renin-Angiotensin-Aldosterone System (RAAS) by inhibiting the Angiotensin-Converting Enzyme (ACE) and further exert their effect on blood pressure controlling [50,51]. Several studies had revealed the protective role of these foods on reducing blood pressure, prolonging life-expectancy and preventing cardiovascular and further diseases [52,53,54]. Furthermore, edible mushrooms had been also reported to be beneficial for regulating intestinal microflora which could also serve as a protective factor for the prevention on a host of diseases [55]. For fresh eggs, there was a study reported a negative association between eggs’ intake and HTN, and a preventive trend was also observed with increase intake of eggs [56]. There is also a meta-analysis showed that one egg per day could potentially decrease the risk of cardiovascular diseases in Asian populations [57], which consisted with current study that a more moderate intake of fresh eggs was observed in the highest quintile.

When considering daily average nutrients’ intakes, we found that participants in highest quintile tend to have less intakes of total energy, fat, carbohydrate, and sodium, but the remains including those we selected as response variables achieved higher intakes compared with other quintile groups. This indicated that adherence to BBP-diet could lead to a higher nutrient density but lower energy diet, which is also consistent with the principles of DASH-diet [42].

The BBP-diet in current study is close to that of other studies. Lee et al. conducted dietary patterns analysis based on population-based cohort study in Shanghai, China, which found that among mid-aged or elder male participants, dietary pattern represented by fruit and dairy had negative correlation with both SBP and DBP, and this effect was more pronounced in those who were heavy alcohol drinkers [58]. Yu et al. conducted a case-control study in Chongqing, China, and found that the pattern, which had higher fish, egg, milk, nut, vegetable, and fruit, but lower salt intake, had strong correlation with lower blood pressure, HTN prevalence, as well as better blood pressure control [59]. A multi-ethnic population-based study in Southwest China showed a greater adherence of grassland pattern (higher consumption of dairy products, soy products, and eggs) was associated with lower risk of HTN [60]. Moreover, a prospective cohort study showed that the Japanese dietary pattern, which had higher consumption of mushrooms and seaweeds, legumes and products, aquatic products, dairy products, eggs, etc., could decrease the risk of CVD mortality; however, a greater adherence of this pattern also showed a higher HTN prevalence and sodium intake [61], which was partly contrary to our findings.

Meanwhile, our research also calculated the DASH-score to test whether our results had conformity with it. The protective effect was observed in the highest quintile group in full-adjustment model. As for the evaluation of DASH-score and well-controlled HTN, a protective effect showed in the highest quintile group. These results indicated that at some point on HTN control and prevention, the BBP-diet has high conformity with DASH-diet. There was still some difference of the dietary portrait between them, since DASH-diet was established in Western populations and may cannot reflect the dietary characteristics of Chinese well. It consisted with the previous study conducted from China Multi-Ethnic Cohort study, DASH score showed an inverse association with HTN, but this dietary pattern may not suit for all the Chinese ethnics based on the culture and other lifestyle factors [18]. Thus, it is essential to discuss in further longitudinal research and find out the dietary pattern which are suitable for blood pressure controlling among Chinese population.

In subgroup analysis, the results showed that protective effect of the BBP-diet on HTN tended to be more pronounced among participants who were male, elderly, and DM patients, and interaction effect modified by DM was also observed. For gender and age difference, previous research showed that premenopausal women had a higher level of cardio-protection than men of similar age, and the former had lower incidence of HTN. The mechanism is involved that sex hormones like estrogen and testosterone could regulate the pathway of vasodilator and vasoconstrictor pathways, including RAAS and the endothelin system [62] and might have stronger effect than diet. However, after postmenopausal, the prevalence of HTN in women would increase due to lose the protective effect of sex hormones [63,64], which to some extent would increase the role of diet. We also observed the interaction effect between BBP-diet score and DM, it might be explained that after being diagnosed with diabetes, patients would autonomously change their diet towards a healthier one which may also further benefit for blood pressure control [65].

Nearly half of Chinese people aged 35–75 years have HTN, but the proportion of treatment and control are still deficient. Plus, with the increasing trend of aging, urbanization, and obesity, there is an urgent need to find a more affordable pathway in blood pressure control [6,66]. Since diet is a key determinant in HTN prevention, treatment, and control, it is important to develop a dietary pattern which is suitable for Chinese people. Meanwhile, people nowadays tend to apply non-pharmacological ways including diets on diseases’ modifications to assist or replace medication treatment [67,68]. Thus, diet intervention may serve as an adjuvant method in the field of HTN and HTN-related diseases prevention and control and has broad prospects in China.

To the best of our knowledge, there is no previous study using RRR method to test blood pressure-related dietary patterns and based on a nationwide Chinese sample. Current study had identified a dietary pattern, which is potentially beneficial for HTN prevention and control, multiple adjustment, and subgroup analysis were conducted which could make our results more plausible. However, several limitations should be noticed in our study. Firstly, because of the cross-sectional design, it is hard to draw a confirmed causality between the dietary pattern and HTN. Secondly, some other food items may be left out due to the limitation of the 64-item food frequency questionnaire. Thirdly, as conducting nationwide research in such a large sample size requires many workforce and material resources, this study might not include all potential variables, especially those laboratory test indexes. Thus, more longitude research and randomized controlled trials are needed in further research.

## 5. Conclusions

Among all the adults participated in CNHS 2015–2017, a BBP-diet identified by RRR method has high conformity with DASH-diet. The current study showed that greater adherence of BBP-diet and DASH-diet was a protective factor for the risk of HTN. Meanwhile, both could also be beneficial for controlling diagnosed HTN. The findings indicated that BBP-diet could serve as a potential method for full cycle prevention and treatment of HTN, which could also be more suitable for the dietary habits of Chinese population.

## Figures and Tables

**Figure 1 nutrients-14-03108-f001:**
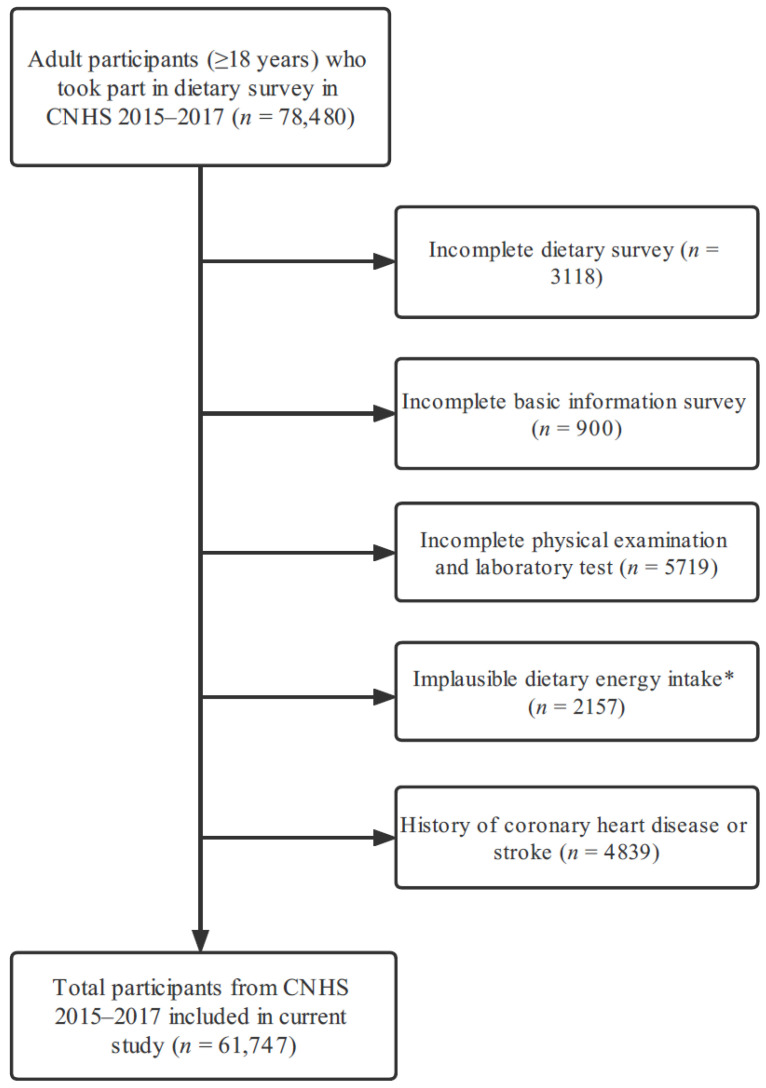
Flow diagram of including eligible participants in current study. * Implausible dietary energy intake was defined as <500 Kcal/day or >5000 Kcal/day.

**Figure 2 nutrients-14-03108-f002:**
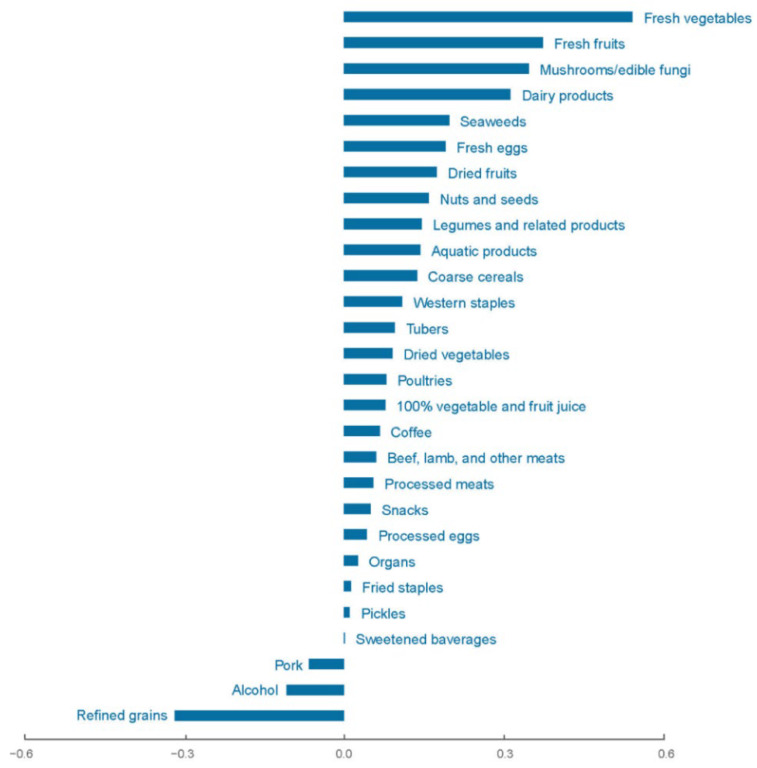
Factor loadings of all food items in BBP-diet.

**Figure 3 nutrients-14-03108-f003:**
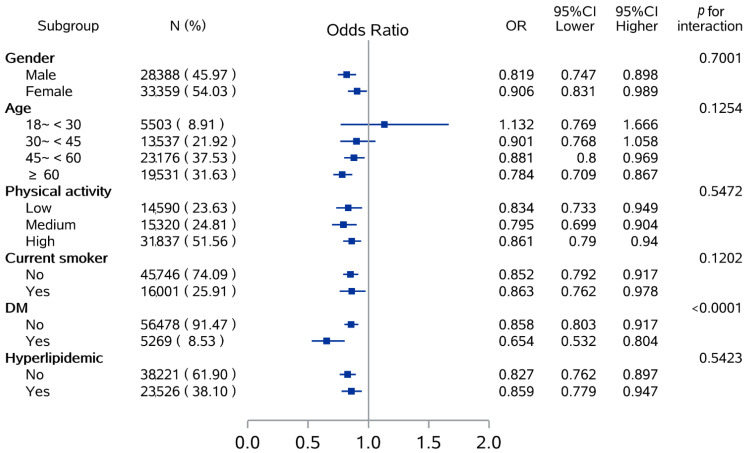
Subgroup analysis of odds ratio (Q5 vs. Q1) between BBP-diet score and risk of HTN according to potential risk factors.

**Table 1 nutrients-14-03108-t001:** Characteristics of participants by quintiles of BBP-diet score.

Variables	Total	Quintile				
		Q1	Q2	Q3	Q4	Q5
Gender *						
Male	28,388 (45.97%)	6498 (52.62%)	6307 (51.07%)	5953 (48.21%)	5338 (43.22%)	4292 (34.76%)
Female	33,359 (54.03%)	5851 (47.38%)	6043 (48.93%)	6396 (51.79%)	7012 (56.78%)	8057 (65.24%)
Age (years) *						
18~<30	5503 (8.91%)	834 (6.75%)	1093 (8.85%)	1185 (9.6%)	1249 (10.11%)	1142 (9.25%)
30~<45	13,537 (21.92%)	2468 (19.99%)	2674 (21.65%)	2799 (22.67%)	2854 (23.11%)	2742 (22.2%)
45~<60	23,176 (37.53%)	4813 (38.97%)	4773 (38.65%)	4653 (37.68%)	4640 (37.57%)	4297 (34.8%)
≥60	19,531 (31.63%)	4234 (34.29%)	3810 (30.85%)	3712 (30.06%)	3607 (29.21%)	4168 (33.75%)
BMI *						
Underweight	2487 (4.03%)	648 (5.25%)	553 (4.48%)	461 (3.73%)	452 (3.66%)	373 (3.02%)
Normal	29,434 (47.67%)	6388 (51.73%)	6182 (50.06%)	5884 (47.65%)	5689 (46.06%)	5291 (42.85%)
Overweight	21,313 (34.52%)	3949 (31.98%)	4089 (33.11%)	4263 (34.52%)	4390 (35.55%)	4622 (37.43%)
Obese	8513 (13.79%)	1364 (11.05%)	1526 (12.36%)	1741 (14.1%)	1819 (14.73%)	2063 (16.71%)
Living area *						
Urban	25,132 (40.7%)	3121 (25.27%)	3838 (31.08%)	4653 (37.68%)	5706 (46.2%)	7814 (63.28%)
Rural	36,615 (59.3%)	9228 (74.73%)	8512 (68.92%)	7696 (62.32%)	6644 (53.8%)	4535 (36.72%)
Education *						
Primary school or below	29,899 (48.42%)	7785 (63.04%)	6786 (54.95%)	6000 (48.59%)	5259 (42.58%)	4069 (32.95%)
Junior middle school	18,945 (30.68%)	3301 (26.73%)	3742 (30.3%)	4054 (32.83%)	4067 (32.93%)	3781 (30.62%)
High school or higher	12,903 (20.9%)	1263 (10.23%)	1822 (14.75%)	2295 (18.58%)	3024 (24.49%)	4499 (36.43%)
Income *						
Not given	9432 (15.28%)	2185 (17.69%)	2146 (17.38%)	1886 (15.27%)	1737 (14.06%)	1478 (11.97%)
Low	16,725 (27.09%)	4248 (34.4%)	3665 (29.68%)	3401 (27.54%)	3103 (25.13%)	2308 (18.69%)
Medium	21,946 (35.54%)	4169 (33.76%)	4437 (35.93%)	4627 (37.47%)	4476 (36.24%)	4237 (34.31%)
High	13,644 (22.1%)	1747 (14.15%)	2102 (17.02%)	2435 (19.72%)	3034 (24.57%)	4326 (35.03%)
Marital status *						
Married	56,609 (91.68%)	11,245 (91.06%)	11,291 (91.43%)	11,378 (92.14%)	11,372 (92.08%)	11,323 (91.69%)
Other status	5138 (8.32%)	1104 (8.94%)	1059 (8.57%)	971 (7.86%)	978 (7.92%)	1026 (8.31%)
Current smoker *						
No	45,746 (74.09%)	8335 (67.5%)	8613 (69.74%)	9005 (72.92%)	9486 (76.81%)	10,307 (83.46%)
Yes	16,001 (25.91%)	4014 (32.5%)	3737 (30.26%)	3344 (27.08%)	2864 (23.19%)	2042 (16.54%)
Second-hand smoking *						
No	31,815 (51.52%)	6079 (49.23%)	5942 (48.11%)	6107 (49.45%)	6487 (52.53%)	7200 (58.3%)
Yes	29,932 (48.48%)	6270 (50.77%)	6408 (51.89%)	6242 (50.55%)	5863 (47.47%)	5149 (41.7%)
Excessive alcohol drinking *						
No	55,451 (89.8%)	10,447 (84.6%)	10,788 (87.35%)	11,081 (89.73%)	11,375 (92.11%)	11,760 (95.23%)
Yes	6296 (10.2%)	1902 (15.4%)	1562 (12.65%)	1268 (10.27%)	975 (7.89%)	589 (4.77%)
Physical activity *						
Low	14,590 (23.63%)	3508 (28.41%)	3004 (24.32%)	2875 (23.28%)	2765 (22.39%)	2438 (19.74%)
Medium	15,320 (24.81%)	2492 (20.18%)	2636 (21.34%)	2999 (24.29%)	3271 (26.49%)	3922 (31.76%)
High	31,837 (51.56%)	6349 (51.41%)	6710 (54.33%)	6475 (52.43%)	6314 (51.13%)	5989 (48.5%)
Sedentary behavior (h) *						
<2	7817 (12.66%)	1908 (15.45%)	1584 (12.83%)	1479 (11.98%)	1483 (12.01%)	1363 (11.04%)
2~3	23,386 (37.87%)	5036 (40.78%)	4908 (39.74%)	4715 (38.18%)	4464 (36.15%)	4263 (34.52%)
≥4	30,544 (49.47%)	5405 (43.77%)	5858 (47.43%)	6155 (49.84%)	6403 (51.85%)	6723 (54.44%)
Sleep duration (h) *						
<7	12,411 (20.1%)	2571 (20.82%)	2389 (19.34%)	2366 (19.16%)	2509 (20.32%)	2576 (20.86%)
7~8	35,926 (58.18%)	6650 (53.85%)	7166 (58.02%)	7291 (59.04%)	7269 (58.86%)	7550 (61.14%)
≥9	13,410 (21.72%)	3128 (25.33%)	2795 (22.63%)	2692 (21.8%)	2572 (20.83%)	2223 (18%)
Medical examination within one year *						
No	46,303 (74.99%)	10,098 (81.77%)	9812 (79.45%)	9427 (76.34%)	8990 (72.79%)	7976 (64.59%)
Yes	15,444 (25.01%)	2251 (18.23%)	2538 (20.55%)	2922 (23.66%)	3360 (27.21%)	4373 (35.41%)
Family history of HTN *						
No	42,328 (68.55%)	9447 (76.5%)	8870 (71.82%)	8527 (69.05%)	8024 (64.97%)	7460 (60.41%)
Yes	19,419 (31.45%)	2902 (23.5%)	3480 (28.18%)	3822 (30.95%)	4326 (35.03%)	4889 (39.59%)
HTN *						
No	37,482 (60.7%)	7224 (58.5%)	7457 (60.38%)	7529 (60.97%)	7700 (62.35%)	7572 (61.32%)
Yes	24,265 (39.3%)	5125 (41.5%)	4893 (39.62%)	4820 (39.03%)	4650 (37.65%)	4777 (38.68%)
DM *						
No	56,478 (91.47%)	11,454 (92.75%)	11,429 (92.54%)	11,306 (91.55%)	11,302 (91.51%)	10,987 (88.97%)
Yes	5269 (8.53%)	895 (7.25%)	921 (7.46%)	1043 (8.45%)	1048 (8.49%)	1362 (11.03%)
Hyperlipidemic *						
No	38,221 (61.9%)	7864 (63.68%)	7831 (63.41%)	7697 (62.33%)	7580 (61.38%)	7249 (58.7%)
Yes	23,526 (38.1%)	4485 (36.32%)	4519 (36.59%)	4652 (37.67%)	4770 (38.62%)	5100 (41.3%)
DASH-score *	24 (22, 27)	20 (18, 22)	23 (20, 25)	24 (22, 26)	26 (23, 28)	28 (26, 31)
SBP *	131.33 (119.67, 146.33)	133 (121, 148.33)	132 (120.33, 147)	131 (120, 146)	130.33 (118.67, 144.67)	130 (118.33, 144.67)
DBP *	78.33 (71.33, 86)	79 (71.67, 86.67)	78.67 (71.67, 86.33)	78.67 (71.67, 86)	78 (71, 85.33)	77.33 (70.33, 84.67)

* Indicates *p* value < 0.05. Values of polytomous variables may not sum to 100% due to rounding. Continuous variables were described as median (P25, P75) due to abnormal distribution and categorical variables were described as amounts with percentages.

**Table 2 nutrients-14-03108-t002:** Association between dietary pattern scores and risk of hypertension in the participants.

Dietary Pattern	Quintile	*N*	No. of Cases	OR (95% CI) *		
				Model I ^†^	Model II ^‡^	Model III ^§^
BBP diet	Q1	12,349	5125	reference	reference	reference
	Q2	12,350	4893	0.925 (0.879, 0.973)	0.957 (0.904, 1.013)	0.968 (0.913, 1.025)
	Q3	12,349	4820	0.902 (0.858, 0.949)	0.915 (0.864, 0.969)	0.935 (0.882, 0.992)
	Q4	12,350	4650	0.851 (0.809, 0.896)	0.860 (0.812, 0.911)	0.885 (0.834, 0.939)
	Q5	12,349	4777	0.889 (0.845, 0.936)	0.791 (0.747, 0.838)	0.842 (0.791, 0.896)
	*p* for trend			<0.0001	<0.0001	<0.0001
DASH diet	Q1	12,298	4673	reference	reference	reference
	Q2	12,843	5023	1.048 (0.996, 1.103)	1.001 (0.945, 1.060)	1.006 (0.949, 1.067)
	Q3	13,487	5277	1.049 (0.997, 1.103)	0.956 (0.903, 1.011)	0.964 (0.909, 1.022)
	Q4	11,257	4450	1.067 (1.012, 1.124)	0.981 (0.924, 1.041)	1.005 (0.945, 1.07)
	Q5	11,862	4842	1.125 (1.069, 1.185)	0.852 (0.803, 0.903)	0.912 (0.854, 0.973)
	*p* for trend			<0.0001	<0.0001	0.0063

* Abbreviations: OR, odds ratio; CI, confidence interval; ^†^ Crude model without adjusting any covariables; ^‡^ Adjusted for age, gender, and BMI; ^§^ Further adjusted for living area, education level, income, marital status, physical activity, sedentary behavior, sleep duration, current smoker (Yes/No), excessive drinking (Yes/No), second-hand smoke (Yes/No), medical examination within one year (Yes/No), family history of HTN (Yes/No), daily energy intake (Kcal/d), and daily sodium intake (mg/d).

**Table 3 nutrients-14-03108-t003:** Association between dietary pattern scores and well-controlled hypertension in the participants.

Dietary Pattern	Quintile	*N*	No. of Well-Controlled	OR (95% CI) *		
				Model I ^†^	Model II ^‡^	Model III ^§^
BBP diet	Q1	1183	241	reference	reference	reference
	Q2	1156	267	0.852 (0.700, 1.037)	0.855 (0.702, 1.043)	0.903 (0.739, 1.104)
	Q3	1276	306	0.811 (0.670, 0.982)	0.79 (0.652, 0.958)	0.89 (0.731, 1.084)
	Q4	1426	360	0.758 (0.630, 0.912)	0.736 (0.611, 0.887)	0.889 (0.732, 1.078)
	Q5	1712	517	0.591 (0.496, 0.705)	0.548 (0.459, 0.654)	0.762 (0.629, 0.924)
	*p* for trend			<0.0001	<0.0001	0.002
DASH diet	Q1	999	191	reference	reference	reference
	Q2	1302	295	0.807 (0.658, 0.990)	0.782 (0.636, 0.960)	0.829 (0.673, 1.021)
	Q3	1402	353	0.703 (0.576, 0.857)	0.701 (0.574, 0.856)	0.795 (0.647, 0.976)
	Q4	1336	345	0.679 (0.556, 0.829)	0.651 (0.532, 0.796)	0.797 (0.646, 0.983)
	Q5	1714	507	0.563 (0.466, 0.680)	0.532 (0.440, 0.644)	0.76 (0.616, 0.938)
	*p* for trend			<0.0001	<0.0001	0.009

* Abbreviations: OR, odds ratio; CI, confidence interval; ^†^ Crude model without adjusting any covariables; ^‡^ Adjusted for age, gender, and BMI; ^§^ Further adjusted for living area, education level, income, marital status, physical activity, sedentary behavior, sleep duration, current smoker (Yes/No), excessive drinking (Yes/No), second-hand smoke (Yes/No), medical examination within one year (Yes/No), family history of HTN (Yes/No), daily energy intake (Kcal/d), and daily sodium intake (mg/d).

## Data Availability

According to the policy of National Institute for Nutrition and Health, China CDC, data related in this research are not allowed to be disclosed.

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
