# Peer review of "Nutrient-Derived Beneficial for Blood Pressure Dietary Pattern Associated with Hypertension Prevention and Control: Based on China Nutrition and Health Surveillance 2015–2017"

_nutrients, 2022, doi:10.3390/nu14153108_

Round 1

Reviewer 1 Report

This study suggested the BBB-diet by analyzing large-scale national survey data for the purpose of providing a beneficial dietary parttern for controlling blood pressure in Chinese people. In order to verify its validity, it was compared and analyzed with the Mediterranean diet or Dash-diet.

Overall, the research purpose and research methods are excellent. However, some typing errors are found in the paper. We also provide some review comments, so I think that if the authors consider these points and revise the manuscript, it will be a better paper..

-       It would be helpful for other researchers to understand the trend of HTN prevalence in China in the introduction of the paper. Also, give a brief introduction to CNHS. This may be described in the Research Methods section.Please briefly present the trends in the prevalence of HTN in China in the introduction.

-       What food items are considered in the final BBB-diet? In the research method session, it was stated that only foods with an absolute factor loading of 0.1 or higher were taken, but meat, tubers, and fermented vegetables seem to have been excluded from supplemenatry table 1. However, in supplemenatry table 2 these food items are included. Please clarify what constitutes the final BBB_diet. It seems to be better if absolute factor loading is indicated next to each food item in Figure 2.

-       line 228-9: Authors described as "As the results shown in Table 1, those participants in the highest quintile 228 had lower systolic and diastolic blood pressure.". However, it seems that many researchers, including myself, would like to see accurate data. Therefore, it is recommended to present the accurate analysis results of SBP and DBP as the average value. Blood pressure is the most important biomedical index in this study.

-       Line 287: It is described as 'HTN patients who received medical treatment within two weeks'. However, I can't find a way to do this, etc. This should be explained in the Research methods session.

-       Several supplementary tables have been presented. Please consider whether the contents corresponding to 3.3 and 3.6 of the research results could not be included in the main text.

The following is about minor issues such as typing errors, etc.

-       Line 67: It is described as CNHS 2015-2017, but the text says that data from CNHS 2015 was obtained. A clear presentation of data set is required.

-       Figure 1. It is suggested that footnotes be added so that the reader can understand the figure alone of Implausible dietary energy intake. (<500kcal/d and >5000kcal/d)

-       Line 122: References to the criteria for hyperlipidemia need to be presented.

-       Line 172: High school -> High school or higher

Author Response

Point 1: It would be helpful for other researchers to understand the trend of HTN prevalence in China in the introduction of the paper. Also, give a brief introduction to CNHS. This may be described in the Research Methods section. Please briefly present the trends in the prevalence of HTN in China in the introduction.

Response 1: Dear reviewer, thank you for your suggestion. We have added the description of the trend of HTN prevalence in China in line 38-39 “In China, it’s estimated that the prevalence of HTN had markly risen from 15.7% in 1991 to 23.3% in 2015 among Chinese adults but remains inadequately controlled”. Meanwhile, for the details of CNHS, the description was also added in line 80-82 “The aim of this survey was to obtain the information of dietary intake, lifestyle habits, medical examination, as well as disease condition and related risk factors among all-age Chinese people…”.

Point 2: What food items are considered in the final BBB-diet? In the research method session, it was stated that only foods with an absolute factor loading of 0.1 or higher were taken, but meat, tubers, and fermented vegetables seem to have been excluded from supplementary table 1. However, in supplementary table 2 these food items are included. Please clarify what constitutes the final BBB-diet. It seems to be better if absolute factor loading is indicated next to each food item in Figure 2.

Response 2: Dear reviewer, thank you very much for your suggestion. The food items we discussed in the part of abstract (line 20-23), results (line 234-237), and discussion (line 331-334) were based on which factor loading was 0.1 or higher, including higher fresh vegetables and fruits, mushrooms/edible fungi, dairy products, seaweeds, fresh eggs, nuts and seeds, legumes and related products, aquatic products, coarse cereals, and less refined grains and alcohol.

As for the main text, tables, and figure, like previous study[1], all the food items included in statistical analysis were presented in figure 1 regardless of their factor loadings to show the general profile of the BBP-diet. And in supplemental table 1, only food items whose factor loading was over 0.1 were displayed to better clarify which food items were used to indicate the characteristics of the BBP-diet. For supplemental table 2, our aim was to try our best to describe all the dietary intakes to help reviewers and readers understand the whole dietary profile in each quintile group.

Point 3: Authors described as "As the results shown in Table 1, those participants in the highest quintile 228 had lower systolic and diastolic blood pressure.". However, it seems that many researchers, including myself, would like to see accurate data. Therefore, it is recommended to present the accurate analysis results of SBP and DBP as the average value. Blood pressure is the most important biomedical index in this study.

Response 3: Dear reviewer, thank you for your suggestion. We have added the median (P25, P75) value of SBP and DBP in each quintile group in Table 1.

Point 4: It is described as 'HTN patients who received medical treatment within two weeks'. However, I can't find a way to do this, etc. This should be explained in the Research methods session.

Response 4: Dear reviewer, sincerely thank you for your suggestion, and we had tried to make it clearer. When we surveyed the part of hypertension in CNHS, we had asked the participants if they had been previously diagnosed with hypertension by a township health center/community health service center/higher-level medical institute, and if they took antihypertensive medication in recent two weeks at the time of survey.

Then, based on the previous research and reports[2,3], in their statement, treatment of hypertension was defined as taking antihypertensive medication currently or within the last 2 weeks as of the time of the interview, further, well-controlled hypertension was considered among patients on treatment if their BP was <140/90mmHg, which is consisted with the definition of well-controlled hypertension in our manuscript. And we have revised the statement and added another more reference in this part.

Point 5: Several supplementary tables have been presented. Please consider whether the contents corresponding to 3.3 and 3.6 of the research results could not be included in the main text.

Response 5: Dear reviewer, thank you for your suggestion. In the part of results in our manuscript, we put the main results in the manuscript, and put other results in the supplementary files to make our manuscript clearer. We also described all the results contained in supplementary files in the main text and reviewers and readers could feel free to check the further information in supplementary files.

Point 6: It is described as CNHS 2015-2017, but the text says that data from CNHS 2015 was obtained. A clear presentation of data set is required.

Response 6: Dear reviewer, thank you for reminding us. All the datasets in our current study were based on CNHS 2015-2017, and we have already corrected this statement (from “CNHS 2015” to “CNHS 2015-2017”) in our title and main text.

Point 7: Figure 1. It is suggested that footnotes be added so that the reader can understand the figure alone of Implausible dietary energy intake. (<500kcal/d and >5000kcal/d)

Response 7: Dear reviewer, thank you for reminding us. And here is the explanation for this point: We have now updated the flow chart (Figure 1) and added the footnote that explains the values of implausible dietary energy intake in line 102. It makes the statement of our manuscript clearer.

Point 8: Line 122: References to the criteria for hyperlipidemia need to be presented.

Response 8: Dear reviewer, thank you for your reminder. This time we have marked the reference in the part of the criteria for hyperlipidemia, further details were shown in our previous article[4].

Point 9: High school -> High school or higher

Response 9: Dear reviewer, thank you for your reminder. We have corrected this part. And we really appreciate that you have proposed the above suggestions and opinions related to our manuscript, and they have really helped us to improve the whole quality of this study. Thank you very much with our best respects!

Reference

  1. Sun, Q.; Wen, Q.; Lyu, J.; Sun, D.; Ma, Y.; Man, S.; Yin, J.; Jin, C.; Tong, M.; Wang, B., et al. Dietary pattern derived by reduced-rank regression and cardiovascular disease: A cross-sectional study. Nutr Metab Cardiovasc Dis 2022, 32, 337-345, doi:10.1016/j.numecd.2021.10.008.
  2. Lao, X.Q.; Xu, Y.J.; Wong, M.C.; Zhang, Y.H.; Ma, W.J.; Xu, X.J.; Cai, Q.M.; Xu, H.F.; Wei, X.L.; Tang, J.L., et al. Hypertension prevalence, awareness, treatment, control and associated factors in a developing southern Chinese population: analysis of serial cross-sectional health survey data 2002-2010. Am J Hypertens 2013, 26, 1335-1345, doi:10.1093/ajh/hpt111.
  3. Chang, J.L.; Wang, Y.; Liang, X.F.; Wu, L.Y.; Ding, G.Q. Report of Chinese Residents' Nutrition and Health Surveillance 2010-2013. Peking University Medical Press: Beijing, China 2016.
  4. Yang, Y.; Piao, W.; Huang, K.; Fang, H.; Ju, L.; Zhao, L.; Yu, D.; Ma, Y. Dietary Pattern Associated with the Risk of Hyperuricemia in Chinese Elderly: Result from China Nutrition and Health Surveillance 2015-2017. Nutrients 2022, 14, doi:10.3390/nu14040844.

Reviewer 2 Report

Thank you for sharing this interesting research which addresses a meaningful topic in the field of nutrition. The manuscript is quite well written and properly organized and contains up-to date bibliography and current discussion. The authors conducted a hard worh and they could be acknowledged for that. They correctly addressed an important nutritional problem, the methods are properly described and discussion is well balanced. The introduction explains and justifies in a coherent and clear manner the background of the study. In my opinion, the article is suitable for publication in the journal although some minor revision is needed. Here I put a few concerns, which in my opinion might improve a quality of the manuscript:

1.      The concept of the methodology and statistical analysis is easy to understood absed on the presented description, however the authors should better clarify some points.

-       It is not fully clear for me why the Authors concentrated on extraction of exactly one dietarry pattern. Potentially, RRR, gives opportunity to extract a few “orthogonal” factors which can together explain much variability in dependent variables, without necessity to restrict consideration only to first the most “powerful”. Could you add an information whether it was checked in some preliminary analysis what rank should be chosen (rank=1) which could help to decide how many DPs should be extracted – that means how many principle components to keep? Or equivalently, was assessed  the effective dimensionality t of the reduced-rank regression firstly? Why not two but only one factor (dietary pattern) was chosen?

-       Subgroup analysis was conducted by comparing the highest quintile group with the lowest in full-adjustment logistic regression – why the analysis were restricted only to comparison of extreme quantile categories? Could you give a rationale/reference from the literature to support such method of data analysis? We can expect the strongest contrast between extreme quintiles, but potentially, some nonlinear pattern of the association is possible in which other than extreme groups differ significantly.

-       In the sentence “However, both DASH and Mediterranean diets were established based on Western populations [????] and may not fit well in representing and measuring Chinese dietary characteristics [14,15]” some references (e.g. two references) to the literature should be added e.g. DOI:https://doi.org/10.1016/j.metabol.2015.02.007

-       The sentence “Indicated by OR values, participants in the highest quintile group of BBP-diet might have less probability for HTN than those in the highest quintile group of DASH-diet”. Please be cautious with drawing such definitive conclusions, not directly and fully confirmed by analysis. Presented ORs are based on different reference groups – or more generally on completely different division into groups.

Please revise article and correct some typos and grammar style: e.g. “Stepwise multiple adjustment logistic regression to compare the effect of dietary patterns on HTN and well controlled HTN” or “As for DASH-score, after being adjusted for all the covariables in model III, negative association between DASH-diet and HTN also showed”

-        The sentence “Therefore, we initially named DP1 as diet beneficial for blood pressure (BBP-diet) for further” should be preceded with the explanation of the notation DP1 – the acronym DP should be introduced firstly (it was in fact not incorporated, I think, at all) for dietary patterns (DP) – line 220, page 6

In summary, the main strengths of this paper are that it addresses an interesting and timely question, there are some limitations, however well described by authors  in discussion section (and typical for any study), it also provides a clear answer.

Author Response

Point 1: It is not fully clear for me why the Authors concentrated on extraction of exactly one dietary pattern. Potentially, RRR, gives opportunity to extract a few “orthogonal” factors which can together explain much variability in dependent variables, without necessity to restrict consideration only to first the most “powerful”. Could you add an information whether it was checked in some preliminary analysis what rank should be chosen (rank=1) which could help to decide how many DPs should be extracted – that means how many principal components to keep? Or equivalently, was assessed the effective dimensionality t of the reduced-rank regression firstly? Why not two but only one factor (dietary pattern) was chosen?

Response 1: Dear reviewer, thank you for your reminding. As previous study, it’s reported that Reduced-Rank Regression (RRR) techniques is applied for derivation of dietary patterns associated with several selected response variables. And by this method, only the first dietary pattern derived from RRR was obtained because it explained the most variation in the responses and accounted for the highest interpretability of findings[1]. Thus, our current study is consisted with the previous research, and kept the first dietary pattern which explained the maximum variation of response variables (66.08%) compared with the remaining dietary patterns extracted by RRR. Therefore, we included the first dietary pattern in further analysis.

Point 2: Subgroup analysis was conducted by comparing the highest quintile group with the lowest in full-adjustment logistic regression – why the analysis was restricted only to comparison of extreme quantile categories? Could you give a rationale/reference from the literature to support such method of data analysis? We can expect the strongest contrast between extreme quintiles, but potentially, some nonlinear pattern of the association is possible in which other than extreme groups differ significantly.

Response 2: Dear reviewer, thank you for your suggestion. We have updated the supplemental table 4 according to your suggestion, and now the OR values and related 95% CI in each quintile group by subgroups are available.

Point 3: In the sentence “However, both DASH and Mediterranean diets were established based on Western populations [????] and may not fit well in representing and measuring Chinese dietary characteristics [14,15]” some references (e.g. two references) to the literature should be added e.g. DOI: https://doi.org/10.1016/j.metabol.2015.02.007

Response 3: Dear reviewer, thank you for reminding us. We have added the relative references to support our main text according to your suggestion[2,3].

Point 4: The sentence “Indicated by OR values, participants in the highest quintile group of BBP-diet might have less probability for HTN than those in the highest quintile group of DASH-diet”. Please be cautious with drawing such definitive conclusions, not directly and fully confirmed by analysis. Presented ORs are based on different reference groups – or more generally on completely different division into groups.

Response 4: Dear reviewer, thank you for reminding us. In the main text of the revised version, we have checked and deleted this statement to let the article more rigorous.

Point 5: Please revise article and correct some typos and grammar style: e.g. “Stepwise multiple adjustment logistic regression to compare the effect of dietary patterns on HTN and well controlled HTN” or “As for DASH-score, after being adjusted for all the covariables in model III, negative association between DASH-diet and HTN also showed”.

Response 5: Dear reviewer, thank you for reminding us. We had revised our article and corrected related typos and grammar style according to your suggestion.

Point 6: The sentence “Therefore, we initially named DP1 as diet beneficial for blood pressure (BBP-diet) for further” should be preceded with the explanation of the notation DP1 – the acronym DP should be introduced firstly (it was in fact not incorporated, I think, at all) for dietary patterns (DP) – line 220, page 6

Response 6: Dear reviewer, thank you for reminding us. We have clarified the DP1 with explanation “the first dietary pattern derived by RRR method” in line 252 to make our manuscript clearer and enhance the readability of this article. We really appreciate your efforts for our article, and we attach great importance to your suggestions on our manuscript. Above suggestions did help us improve the quality of our work. Thank you very much with our best respects!

Reference

  1. Lazarova, S.V.; Jessri, M. Associations between dietary patterns and cardiovascular disease risk in Canadian adults: a comparison of partial least squares, reduced rank regression and the simplified dietary pattern technique. Am J Clin Nutr 2022, 10.1093/ajcn/nqac117, doi:10.1093/ajcn/nqac117.
  2. Ozemek, C.; Laddu, D.R.; Arena, R.; Lavie, C.J. The role of diet for prevention and management of hypertension. Curr Opin Cardiol 2018, 33, 388-393, doi:10.1097/HCO.0000000000000532.
  3. Grosso, G.; Stepaniak, U.; Micek, A.; Topor-Madry, R.; Stefler, D.; Szafraniec, K.; Bobak, M.; Pajak, A. A Mediterranean-type diet is associated with better metabolic profile in urban Polish adults: Results from the HAPIEE study. Metabolism 2015, 64, 738-746, doi:10.1016/j.metabol.2015.02.007.
